# Health System Barriers and Facilitators to Delivering Additional Vaccines through the National Immunisation Programme in China: A Qualitative Study of Provider and Service-User Perspectives

**DOI:** 10.3390/vaccines9050476

**Published:** 2021-05-08

**Authors:** Dan Gong, Qiyun Jiang, Tracey Chantler, Fiona Yueqian Sun, Jiatong Zou, Jiejie Cheng, Yuqian Chen, Chengyue Li, Mei Sun, Natasha Howard

**Affiliations:** 1Department of Health Policy and Management, School of Public Health, Fudan University, 130 Dong’an Road, Shanghai 200032, China; 18211020057@fudan.edu.cn (D.G.); qiyun_jiang@163.com (Q.J.); 18211020050@fudan.edu.cn (J.C.); chenyuqian96@163.com (Y.C.); 2Research Institute of Health Development Strategies, Fudan University, 130 Dong’an Road, Shanghai 200032, China; 3Department of Global Health and Development, London School of Hygiene & Tropical Medicine, 15–17 Tavistock Place, London WC1H 9SH, UK; Tracey.Chantler@lshtm.ac.uk (T.C.); natasha.howard@nus.edu.sg (N.H.); 4Department of Infectious Disease Epidemiology, London School of Hygiene & Tropical Medicine, University of London, Keppel Street, London WC1E 7HT, UK; Fiona.Sun@lshtm.ac.uk; 5Department of Organization and Personnel, Shanghai Municipal Center for Disease Control and Prevention, 1380 West Zhongshan Road, Shanghai 200336, China; zoujiatong@scdc.sh.cn; 6Saw Swee Hock School of Public Health, National University of Singapore, 12 Science Drive 2, Singapore 117549, Singapore

**Keywords:** vaccination, routine immunization, health systems, barriers and facilitators, China

## Abstract

In China, there are two categories of vaccines available from the Chinese Center for Disease Control and associated public health agencies. Extended Program of Immunization (EPI) vaccines are government-funded and non-EPI vaccines are voluntary and paid for out-of-pocket. The government plans to transition some non-EPI vaccines to EPI in the coming years, which may burden public health system capacity, particularly in terms of budget, workforce, supply chains, and information systems. Our study explored vaccinator and caregiver perspectives on introducing non-EPI vaccines into routine immunization and perceived facilitators and barriers affecting this transition. We conducted a qualitative study from a realist perspective, analysing semi-structured interviews with 26 vaccination providers and 160 caregivers in three provinces, selected to represent regional socioeconomic disparities across Eastern, Central, and Western China. Data were analysed thematically, using deductive and inductive coding. Most participants were positive about adding vaccines to the national schedule. Candidate EPI vaccines most frequently recommended by participants were varicella, mumps vaccine, and hand–foot–mouth disease. Providers generally considered existing workspaces, cold-chain equipment, and funding sufficient, but described frontline staffing and vaccine information systems as requiring improvement. This is the first qualitative study to explore interest, barriers, and facilitators related to adding vaccines to China’s national schedule from provider and caregiver perspectives. Findings can inform government efforts to introduce additional vaccines, by including efforts to retain and recruit vaccine programme staff and implement whole-process data management and health information systems that allow unified nationwide data collection and sharing.

## 1. Introduction

The Chinese government has promoted childhood vaccination since the 1950s [1], adopting a “prevention first” principle into health policy and implementing countrywide vaccination against vaccinia (smallpox). In 1978, China initiated its Expanded Program on Immunization (EPI) [2], providing Bacillus Calmette-Guerin (BCG), Diphtheria, Pertussis, Tetanus (DPT), polio, and measles vaccines as “4 vaccines against 6 vaccine-preventable diseases” [3]. In 2007, the government expanded national provision to “14 vaccines against 15 vaccine-preventable diseases” [4]. It is now considering incorporating additional vaccines, including the hand, foot, and mouth disease (EV71) vaccine and the pneumococcal conjugate vaccine (PCV).

The Chinese public health system, responsible for immunization, is a five-tiered system from central to township level. It consists primarily of the National Health Commission (China’s Ministry of Health equivalent) and subordinate bodies, the Chinese Center for Disease Control and Prevention (CDC) and subordinate bodies, township health centres in rural areas, and community health centres in urban areas. The system currently delivers two types of vaccines; EPI vaccines are paid for by the fiscal budget and non-EPI vaccines are voluntary and paid for out-of-pocket by recipients. EPI and non-EPI vaccines are both distributed through public vaccine supply chains run by the Chinese CDC and its subordinate bodies. Appendix A shows vaccines provided in China as EPI or non-EPI. The government’s plans to transition some non-EPI vaccines to the EPI schedule in the next few years will increase the burden on the limited public health system capacity, particularly in terms of budget, workforce, supply chains, and information systems.

Research on the potential effects of expanding China’s EPI schedule includes studies on the effects of introducing MMR and hepatitis A vaccines on mumps [5] and hepatitis A antibody levels [6], respectively, influencing factors, such as disease burden [7], cost-effectiveness of vaccine introduction [8,9], vaccine safety and effectiveness [10,11], and international or national advocacy [12]. Other studies investigate cold chain equipment [13] and personnel [14] capacities. However, authors found no qualitative studies on provider or service-user perspectives of introducing non-EPI vaccines into the routine schedule in China. Research in other countries has included providers’ attitudes and how such considerations hindered the introduction of the dengue vaccine in Colombia and Venezuela [15]. Thus, it is important to examine these perspectives on the capacity of the Chinese public health system to deliver additional vaccines, while maintaining current performance levels. Our study aimed to examine vaccinator and caregiver attitudes to introducing non-EPI vaccines into routine immunization and perceived facilitators and barriers affecting this transition.

## 2. Materials and Methods

### 2.1. Study Design

We adopted a qualitative study design, using a realist approach as described by Maxwell [16], which included semi-structured interviews with vaccination providers and caregivers in Guangdong, Henan, and Sichuan provinces. Realist approaches are common, though often not explicit, in qualitative public health research, as they enable pragmatic engagement with a ‘real world’, to which our concepts and theories refer, which remains a powerful and resilient research stance in public health.

### 2.2. Research Question

Our primary research question was “What are vaccination provider perspectives on the Chinese public health system capacity to deliver additional vaccines, such as EV71 vaccine and PCV?”

### 2.3. Study Sites

First, we purposively selected three provinces to represent regional socioeconomic disparities across Eastern, Central, and Western China that could influence public health system capacity to deliver immunization services (see Appendix A). Second, we randomly selected 1 rural county and 1 urban district per province to account for China’s dichotomous rural–urban service delivery structure. Third, we purposively selected 3–5 community healthcare centres (i.e., vaccination facilities) in each of the 6 selected districts/counties, and 1 local CDC in each province. Thus, 22 vaccination facilities and 3 district/county CDCs were included in this study.

### 2.4. Participant Recruitment

To obtain a range of perspectives, we purposively sampled and recruited an EPI manager or senior immunization staff at each of the 22 vaccination facilities and 3 district/county CDCs. At each vaccination facility, we similarly purposively recruited approximately 5 caregivers in waiting rooms after their children were vaccinated. To account for non-attendees, we additionally recruited approximately 3 caregivers per facility catchment community, through a random sample of the vaccination register system (i.e., Immunization Information System). This system includes information on children’s socio-demographics (i.e., name, birthdate, birthplace), immunization record (i.e., type, dose, date), and caregiver information (i.e., name, phone number) for use in sending vaccination reminders. In total, we interviewed 26 providers and 160 caregivers.

### 2.5. Data Collection

We conducted interviews April–July 2019, using interview guides pre-tested in Shanghai (i.e., at 1 rural and 1 urban public facility, 2 corresponding district/county CDCs, and Shanghai CDC). Provider topics included interest in and feasibility of including new vaccines, challenges, and enablers of converting vaccines from private-sector to public-sector delivery, and vaccination experiences. Caregiver topics included interest in additional public vaccines, any known barriers and enablers, and vaccination experiences.

All interviews were conducted face-to-face in Mandarin by one of 7 trained interviewers, after written informed consent was obtained. Consent consisted of an interviewer introduction to the purpose, aims, topics, and ethics of the study, discussing participation, and answering any questions. Provider interviews lasted approximately 45 min and caregiver interviews lasted about 25 min. Interviews were audio recorded and investigators also compiled field notes on the interview environment, interviewee body language, self-reflection, and other relevant information, which was used to inform analysis. As caregivers did not contribute to barriers or facilitators, their views are only reported under willingness and acceptability.

### 2.6. Analysis

We transcribed audio files verbatim in Mandarin and entered transcripts into the NVivo 11 (QSR International Pty Ltd., Victoria, Australia) data management software and used Braun and Clarke’s six-stage thematic analysis approach [17]. First, three investigators (i.e., MS, DG, JC) read and became familiarised with the data. Second, investigators independently generated initial codes. Third, investigators developed a coding structure iteratively, collating codes related to interest, barriers, and facilitators into preliminary themes, which was translated into English for discussion with non-Mandarin fluent co-authors. Investigators examined relationships between codes, focusing on the research question, compiled them, and summarised the content of each theme. This summary was translated into English for inputs from non-Mandarin fluent co-authors. Fourth, investigators conducted thematic mapping. Fifth, investigators refined and defined independent themes through discussion and further integration. Finally, all investigators reviewed and refined themes during the reporting process. To protect participant privacy and anonymity, we assigned identification codes (i.e., “P#” for providers, “C#” for caregivers).

## 3. Results

### 3.1. Participant Characteristics

Table 1 provides the socio-demographic characteristics of participants. The 26 providers (i.e., 18 EPI managers and 5 vaccinators from vaccination facilities, 3 CDC EPI managers) were from Sichuan (38.5%), Henan (34.6%), and Guangdong (26.9%) provinces. The average age was 42.5 and approximately 46.2% had vocational college training or below, mainly in nursing (42.3%) and preventive medicine (23.1%). Of the 160 caregivers, we interviewed 94 at vaccination facilities and 66 through vaccination registers. Caregivers were from Sichuan (26.9%), Henan (31.9%), and Guangdong (41.2%) provinces. Approximately 79.4% were women, and all were parents of a vaccinated child. The average age was 35 years old and 15.0% had vocational college training or above (vocational college provides post-secondary vocational training rather than a bachelor’s degree).

### 3.2. Analytical Themes

We describe findings below under willingness and acceptability, barriers (i.e., workforce and workload; information systems), and facilitators (i.e., workspace and equipment; funding; previous experience).

#### 3.2.1. Willingness and Acceptability

##### Providers

Most participants said they welcomed moves to transition non-EPI vaccines into the national schedule. Primarily, they suggested it could reduce the burden of explaining non-EPI vaccines to caregivers. As non-EPI vaccines required payment and were sometimes very expensive, parents consulted providers about vaccine efficacy, safety, and price before vaccination. Secondly, vaccine availability through EPI would attract more parents and vaccination rates would likely improve. As one vaccinator noted:

“We certainly welcome it! We could reduce the communication time with parents and would not need to emphasize the cost, so the burden would be much smaller.” (P8)

The candidate EPI vaccines most frequently recommended by participants were the varicella vaccine, the mumps vaccine, the hand, foot and mouth disease (HFMD) vaccine, the influenza vaccine, and the 13-valent pneumonia vaccine (while MMR vaccination was provided through EPI, a mumps-only vaccination was not). The first three were well-known and commonly requested, as these viruses could cause epidemics through public interactions in social places such as schools, while the latter two prevented seasonal infections such as influenza. As one EPI manager suggested:

“Chickenpox, mumps [...]. These are infectious diseases. There are many children in the school. Once there is an outbreak, it will be very serious. So, I think these two vaccines are very important.” (P11)

##### Caregivers

Most caregivers said that they would support the transition of non-EPI vaccines to EPI delivery, primarily because it would reduce their financial burden and because they suggested that the safety of EPI vaccines was better guaranteed. Caregivers suggested that they would be willing to accept longer waiting times if there was an increase in the number of people attending facilities for vaccination. As one caregiver noted:

“Maybe after some [non-EPI] vaccines become free, more people would come to vaccinate, and the waiting time would also become longer, but it is acceptable.” (C21)

Most caregivers reported that their children had received non-EPI out-of-pocket vaccines, indicating both a strong demand for childhood vaccinations and a presumption that EPI vaccines were insufficient. The three preferred non-EPI vaccines recommended by caregivers for transition were for chickenpox, HFMD, and mumps, reportedly because children were particularly vulnerable to these diseases. As one caregiver noted:

“I think chickenpox vaccine is necessary, because every child may experience it. [If infected] the child will be very uncomfortable and it will be troublesome to deal with it. And I also hope the flu vaccine [can become an EPI vaccine]...”(C32)

#### 3.2.2. Workforce and Workload Barriers

Most providers suggested that vaccination staff numbers were insufficient, especially in regard to young people with public health training and staff with vaccination licenses. They noted it was difficult to recruit people to prevention and healthcare departments, due to low wages and the perceived low value of the positions. Most suggested more staff would be needed if additional vaccines were added to the national schedule. As a provider from Guangdong province observed:

“There are 13 staff in our department, only three of them are full-time in immunization planning. Others have part-time jobs, such as prevention and control of infectious diseases and chronic diseases, and health supervision and management. So, the lack of staff is a big problem. Moreover, many young staff are temporary workers and it is very difficult to become regular employees, which led to their resignations.”(P20)

Some providers suggested that an increase in the rotation of existing staff could solve staff shortages. As a provider from Henan explained, when asked about additional vaccines:

“There is no need to increase the number of staff. The four of us could take turns to arrange shifts and complete the work.” (P5)

As vaccination facilities had different vaccinator numbers, workspaces, and cold storage capacities, daily vaccination volumes in each facility varied between approximately 50 and 270 injections. Most providers indicated that an increase in vaccination volume of 10–50% would be acceptable, considering current space, staff, and cold storage capacities. However, if vaccination volumes doubled, vaccination rooms would become crowded, while staff and cold storage capacity would be insufficient. A few providers in Guangdong province suggested current vaccination volumes were at upper limits:

“270 injections a day has been a lot [...]. Maybe it can be increased by about 10% to about 50%, and no more can be sustained.” (P22)

#### 3.2.3. Information System Barriers

Most providers suggested that the current vaccination information system was imperfect. For example, electronic calling was not popular and information systems were not interconnected nationwide. One provider noted particular challenges relating to mobility:

“Immunization Information Systems can be connected within the city, but not within the province, and not among the provinces. There are so many migrant people in our district. It’s a problem because migrant people have been vaccinated in other places, but you can’t download this information when he comes here.” (P20)

Appointment setting was primarily traditional, using reservation books and oral notification. Few providers used apps, such as “Xiao Doumiao” and WeChat official, due to insufficient promotion and glitches in service functions. As a facility manager observed:

“Very few people use it [apps] now. We used to teach caregivers, but now we can’t teach too many people. Because we are only four people. We are too busy.” (P25)

#### 3.2.4. Workspace and Equipment as Facilitators

Most providers indicated they had sufficient workspace, though a few said their workspace did not meet standards and some did not have observation space for adverse events following immunization (AEFIs). One Guangdong provider described:

“[Facilities in] Guangdong Province are divided by stars, from one-star to five-star. We can only achieve three-star. Some infrastructure are not up to standard. For example, there is no special room for AEFIs.” (P20)

Providers suggested that current space was sufficient if vaccination volume increased by 30–50%; for example, if 1–2 non-EPI vaccines were transitioned to the national schedule. However, if vaccination volumes more than doubled, they would not have capacity. As a Henan provider observed:

“At present—when there are 60–70 recipients per day—there will be no crowding, because everyone conscientiously queues up. But if a new EPI vaccine is added, the space will start to crowd at 100 recipients per day.” (P1)

Most providers reported their current cold storage capacity as sufficient, with some spare that could accommodate 1–2 new EPI vaccines. Only three providers indicated no spare cold storage capacity at their facility. If non-EPI vaccines transferred to EPI vaccines, others suggested that, either way, they could apply for additional cold storage equipment:

“It [cold storage space] could be enough. If not, we can apply for new refrigerators from the department of health...” (P9)

#### 3.2.5. Funding as Facilitator

Current national vaccination schedule funding is allocated by the government, with only a few providers suggesting it was insufficient or difficult to apply for electronic equipment upgrades. As a facility manager from Guangdong said:

“The funds are provided by the district government and our institutional funds are sufficient. All the money we collect will be handed over to the district government, which will then allocate us funds.” (P21)

Most providers indicated that adding additional vaccines to the national schedule would likely have little effect on their facility’s capital allocation or salaries, though they hoped it would increase salary funding.

“We only provide vaccination services. As for funding, the government will guarantee it.” (P17)

#### 3.2.6. Previous Experience as Facilitator

All providers said their experience of the polio vaccine changing from oral to injection on the national schedule had minimal impact on their workload. Several suggested that it was more convenient to provide injected than oral vaccines to children. As a Guangdong provider explained:

“In fact, I would rather offer injections than ‘sugar pills’. Because some babies are too young, I need to prepare water cups and boiling water, and use a spoon to break the ‘sugar pills’ before feeding them, which is too troublesome!” (P26)

## 4. Discussion

This qualitative study explored providers’ interest, barriers, and facilitators to, and caregivers’ willingness and acceptability of, delivering additional vaccines through the national EPI schedule. Such frontline perspectives are important and can help inform policy development. Our study indicated providers and caregivers were positive about including additional vaccines in the national schedule, and providers expected existing workspaces, cold chain equipment, and funding could meet demands for 1–2 additional EPI vaccines, which differs from findings in other middle-income countries where financing is a major barrier [15,18]. Several possible explanations for these differences exist. First, the Chinese government has routinely prioritised the national immunization programme and allocated it specific funding (e.g., vaccination funding per person was ensured through the 2009 Essential Public Health Services guidance) [19]. Second, the government invested significantly in infectious disease control through the CDC system, including workspaces and cold chain equipment at vaccination facilities, after the 2002–2004 SARS crisis [20]. Third, the 2005 Regulation on the Administration of Circulation and Vaccination of Vaccines [21], which specifies standards for vaccination procedures, workspaces, equipment, and facilities, is strictly regulated. This enabled major improvements in vaccination facilities and provision, creating a strong foundation on which to introduce additional vaccines. Fourth, the government has focused on training vaccination providers on vaccine administration policies and procedures, through both domestic and international sessions [22,23,24], which may have enhanced their interest in delivering additional vaccines through the national schedule.

The main potential barriers described by providers were personnel shortages and health information system weaknesses [25,26]. Experiences from HPV vaccine introductions in low- and middle-income countries also indicated that insufficient infrastructure (e.g., electronic databases) and human resources were the main barriers [27], though a study of Hepatitis B vaccine introduction in Ethiopia did not describe human resources as a challenge [28]. In China, increased public demand for vaccination, including self-funded non-EPI vaccines, has increased vaccination facility workloads. At the same time, some EPI staff have taken on additional responsibilities such as chronic and infectious disease management [29,30]. Limited EPI salaries and promotion opportunities have reduced its attractiveness as a career choice and increased vaccination facility staffing instability and turnover [31]. Increasing vaccination workload by introducing additional vaccines, without addressing staffing constraints, could lead to more serious personnel shortages. Therefore, the government should initiate efforts to both retain and recruit EPI staff, using financial and nonfinancial incentives (e.g., increased salaries, continuing education, career development) [29,32].

Inadequate facility-level immunization information systems could affect the transition of additional vaccines into the national schedule. Although China is promoting the construction of Immunization Information Systems (IISs) [33], several problems remain. First, the information system does not effectively address whole-process vaccination management of appointments, registration, and AEFI recording. Second, most immunization information cannot be shared between facilities, cities, and provinces [34], thus exacerbating delayed/missed and repeated vaccinations, particularly among migrant children [35]. To ensure additional vaccines do not create overwhelming EPI workloads and caregiver confusion, the government must ensure IISs provide whole-process services and whole-link management [36,37], with unified data collection and sharing standards that allow cross-regional data exchange [38].

### Limitations

Several study limitations should be considered. First, due to budget and time constraints, we included three representative provinces and findings should not be generalised to all provinces or districts in China. Second, while we included a range of provider levels, we did not interview other key stakeholders, such as providers in other departments, other ministries (e.g., finance), or vaccine manufacturers, who would likely have provided different and additional perspectives.

## 5. Conclusions

This was the first qualitative study, of which authors are aware, to explore willingness, barriers, and facilitators related to adding vaccines to China’s national schedule from providers’ and caretakers’ perspectives. Most providers and caregivers supported the transition of non-EPI vaccines to EPI delivery. We found that, based on current workloads and previous experience, most providers were positive about adding new vaccines to the national schedule and considered existing workspaces, cold-chain equipment, and funding sufficient to meet the demands of 1–2 additional vaccines. However, frontline staffing and information systems required improvement. Improvements, which could be implemented before or alongside vaccine introductions, should include efforts to retain and recruit EPI staff and implement a whole-process/whole-link health information system that allows unified nationwide data collection and sharing standards.

## Figures and Tables

**Table 1 vaccines-09-00476-t001:** Participant characteristics.

Provider Characteristics	*n* (%) or Mean ± SD
Province	
Guangdong	7 (26.9)
Henan	9 (34.6)
Sichuan	10 (38.5)
Age in years [mean]	42.5 ± 7.3
Age range	
<35	5 (19.2)
35–45	12 (46.2)
>45	9 (34.6)
Gender	
Male	9 (34.6)
Female	17 (65.4)
Educational level	
Below vocational college	6 (23.1)
Vocational college	6 (23.1)
Undergraduate	5 (19.2)
Masters	1 (3.8)
Missing	8 (30.8) *
Major	
Clinical medicine	5 (19.2)
Nursing	11 (42.3)
Preventive medicine	6 (23.1)
Sanitary inspection	1 (3.8)
Chinese medicine	1 (3.8)
Missing	2 (7.7) *
Personnel category	
CDC EPI manager	3 (11.5)
EPI manager	18 (69.3)
Vaccinator	5 (19.2)
**Caregiver Characteristics**	***n* (%) or Mean ± SD**
Province	
Guangdong	66 (41.2)
Henan	51 (31.9)
Sichuan	43 (26.9)
Age in years [mean]	35.0 ± 11.0
Age range	
<35	100 (62.5)
35–45	19 (11.9)
>45	30 (18.8)
Missing	11 (6.8) *
Gender	
Male	33 (20.6)
Female	127 (79.4)
Educational level	
Below vocational college	34 (21.3)
Vocational college	10 (6.3)
Undergraduate	13 (8.1)
Master	1 (0.6)
Missing	102 (63.7) *
Source of respondents	
Vaccination facilities	94 (58.8)
Vaccination registers	66 (41.2)
Regional	
Urban	81 (50.6)
Rural	79 (49.4)

Note: * Some participants were unwilling to disclose personal information.

## Data Availability

The data presented in this study are available on request from the corresponding author.

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
