# Peer review of "Health System Barriers and Facilitators to Delivering Additional Vaccines through the National Immunisation Programme in China: A Qualitative Study of Provider and Service-User Perspectives"

_vaccines, 2021, doi:10.3390/vaccines9050476_

Round 1

Reviewer 1 Report

Overall this is an interesting paper and an impressive effort to interview 160 caregivers. Front line perspectives on immunisation providers are useful ways to evaluate immunisation programmes. In it's current form however it is not suitable for publication.

Overall points:

I am not familiar with the Chinese health system nor the immunisation programme there so I think for a non-familiar reader there are improvements that could be made to add more context.

The discussion, limitations and conclusion needs major revisions - in particular I am very concerned by the lack of attention to detail prior to submission. If this was indeed reviewed by all authors prior to submission how was the generic content 'author instructions' provided by the journal missed and included in the final submission - see lines 309-312?

Given that 160 caregivers were interviewed I would have liked more information both on their characteristics (a table would be helpful) and also more information and discussion on how their responses differed.

These are my specific comments for improvement:

  •  Add more explanation of the Chinese information program - for example in the methods section vaccination registers are mentioned but these had not been previously explained?
  • What is the international equivalent of vocational college? High School?
  • Results - please provide a map of where the interviews took place maybe or explain in more detail the socio-economic demographics of the provinces where the interviews were conducted. Urban versus rural not detailed enough.
  • The first section of the results (willingness) discusses that Mumps was one of the vaccines participants wanted adding but this is in the EPI included section of the table - this leaves the reader confused. Similarly this same reference to Mumps is made in the discussion.
  • Caregivers - I would like much more information on their responses in the findings and also the discussion. Maybe this could be more clearly presented.
  • What is the volume of vaccinations each healthcare facility delivered each day normally - this would aid framing of the additional workload discussions.
  • The interview guides need including please - both providers and caregivers.
  • There is a passing reference to other countries literature in the introduction but a far better and constructive discussion including reference to previous literature and other countries relevant experiences is needed in the discussion.

Reviewer 2 Report

The study is interesting, the text is well structured and understandable. The main elements are present. I have some comments on the presentation of the article to make it more readable for readers. My comments are included directly in the PDF.  

Round 2

Reviewer 1 Report

Thank you for taking on board the comments - this is a lot clearer to read now - particularly appreciate the addition of table 1. It maybe needs to be made much clearer earlier on in the study that caregivers responses with regards to 'willingness and acceptability' of additional vaccines being provided only were analysed. Are the 160 caregivers wider responses being written up elsewhere as a study?

Minor suggestions below.

ABSTRACT

  • The addition of the first sentence to the abstract is welcome. I would rephrase it though so it reads more concisely. For example: In China there are two categories of vaccines available from the Chinese Center for Disease Control and associated public health agencies.  Category 1 (Extended Program Immunisation EPI) vaccines are government funded and Category 2 vaccines are voluntary and paid for out-of-pocket.  
  • EPI and category 1 are used interchangeably throughout the paper - it is confusing at times. From my reading of the paper category 1 and EPI vaccines are the same thing - needs consistency throughout maybe. For example this sentence in the abstract needs to explain that the study was looking at more category 2 vaccines becoming category 1 vaccines as that is what the abstract starts out with. To then introduce EPI vaccines as a way of describing category 1 vaccines is confusing.: Our study explored vaccinator and caregiver perspectives on introducing non-EPI  (Expanded Program on Immunization) vaccines into routine immunization and perceived facilitators and barriers affecting this transition. 

INTRODUCTION

  • Thank you for expanding on the background - this is helpful. Minor point but this sentence is too long and uses 'subordinate' too many times. Suggest rephrase/shorten. The Chinese public health system responsible for immunization consists of the Na- 56 tional Health Commission (China’s Ministry of Health equivalent) and its subordinate 57 bodies, the Chinese Center for Disease Control and Prevention (CDC) and its subordinate 58 bodies, township health centers in rural areas, community health centers in urban areas, 59 and other immunization facilities at different administrative levels and categories, which 60 together form a 5-tiered system from central to township level.

METHODS

  • Reads well and processes clearly described. I think need to explain to the reader more clearly (and the interview questions supplied help this) that the caregivers perspectives were only sought with regards to willingness and acceptability. For example could expand and adapt to include this sentence from the results section in the methods section: 'As caregivers did not provide additional contributions on 161 barriers or facilitators, their views are only reported under willingness and acceptability.'

RESULTS

  • In the results section on willingness and acceptability (3.2.1) could add  subheadings outlining 'providers' results and then 'caregivers' results so it is easier to read.
  • There are no caregiver quotes included? Suggest adding given 160 interviews were undertaken there must be some rich data?

DISCUSSION

  • Again as mentioned before need to be clear that study explored in detail providers thoughts on barriers and facilitators to adding in additional vaccines to category 1 but only caregivers willingness and acceptability.  Maybe rephrase the first sentence of discussion.
  • On the same theme make sure it is clear you are describing what 'providers' said. eg line 292: The main potential barriers described were personnel shortages and health information system weaknesses [25, 26]. Maybe add in 'by providers'.
  • Add in line on caregivers willingness findings to conclusion
